# Macroscopic resting state model predicts theta burst stimulation response: A randomized trial

**Neda Kaboodvand**[1,2,3☉], **Behzad Iravani**[1,2☉]*, **Martijn P. van den Heuvel**[4], **Jonas Persson**[3], **Robert Boden**[3]

**1** Department of Neurology and Neurological Sciences, Stanford University, Stanford, California, United States of America, **2** Department of Clinical Neuroscience, Karolinska Institutet, Stockholm, Sweden, **3** Department of Medical Sciences, Psychiatry, Uppsala University, Uppsala, Sweden, **4** Department of Complex Traits Genetics, Center for Neurogenomics and Cognitive Research, VU Amsterdam, Amsterdam, The Netherlands

☉ These authors contributed equally to this work.
* behzad.iravani@ki.se

**Data Availability Statement:** The structural data used for creating the computational models is openly available at http://www.humanconnectomeproject.org/data/. The functional

## Abstract

Repetitive transcranial magnetic stimulation (rTMS) is a promising alternative therapy for treatment-resistant depression, although its limited remission rate indicates room for improvement. As depression is a phenomenological construction, the biological heterogeneity within this syndrome needs to be considered to improve the existing therapies. Whole-brain modeling provides an integrative multi-modal framework for capturing disease heterogeneity in a holistic manner.

Computational modelling combined with probabilistic nonparametric fitting was applied to the resting-state fMRI data from 42 patients (21 women), to parametrize baseline brain dynamics in depression. All patients were randomly assigned to two treatment groups, namely active (i.e., rTMS, n = 22) or sham (n = 20). The active treatment group received rTMS treatment with an accelerated intermittent theta burst protocol over the dorsomedial prefrontal cortex. The sham treatment group underwent the identical procedure but with the magnetically shielded side of the coil.

We stratified the depression sample into distinct covert subtypes based on their baseline attractor dynamics captured by different model parameters. Notably, the two detected depression subtypes exhibited different phenotypic behaviors at baseline. Our stratification could predict the diverse response to the active treatment that could not be explained by the sham treatment. Critically, we further found that one group exhibited more distinct improvement in certain affective and negative symptoms. The subgroup of patients with higher responsiveness to treatment exhibited blunted frequency dynamics for intrinsic activity at baseline, as indexed by lower global metastability and synchrony.

Our findings suggested that whole-brain modeling of intrinsic dynamics may constitute a determinant for stratifying patients into treatment groups and bringing us closer towards precision medicine.

scans and behavioral data cannot be publicly shared due to institutional and ethical constraints. We have restrictions regarding the processing of personal data. The research participants have not consented to public sharing of pseudonymized data, and thus we do not have ethical permission to openly share these data. Neither do we have a legal permission according to the General Data Protection Regulation (GDPR) regarding classified personal data. All that being said, the functional data are locally available by contacting the third-party contact point: dataoffice@uu.se. The code for the whole-brain modeling is publicly available at https://github.com/Behzad-Iravani/WBM-TMS-and-Depression-.

**Funding:** N.K, B.I. and R.B. were financially supported by the Swedish Research Council (https://www.vr.se/), with the grant numbers: 2020-00724, 2021-06645 and 2016-02362. Data preprocessing was enabled by resources provided by the Swedish National Infrastructure for Computing (SNIC) at Uppsala Multidisciplinary Center for Advanced Computational Science (UPPMAX) partially funded by the Swedish Research Council through grant agreement no. 2018-05973. The funder had no role in study design, data collection and analysis, decision to publish, or preparation of the manuscript.

**Competing interests:** The authors have declared that no competing interests exist.

## Author summary

There are multiple therapeutic protocols with a limited remission rate, for treatment-resistant depression, using repetitive transcranial magnetic stimulation (rTMS). It is still unclear how we match different rTMS protocols to patients to optimize the therapy. Currently, the process of determining the best rTMS protocol for each individual involves trial and error. Whole-brain computational modelling paves the way to find the optimal therapeutic protocol for each patient, by integrating multi-modal neuroimaging through theoretical models of brain dynamics. In this work, whole-brain modelling helped us identify two covert clinically relevant subtypes in our depression cohort, exhibiting different responses to the same rTMS therapy applied over the dorsomedial prefrontal cortex. Patients who were assigned to the subtype with blunted resting-state frequency dynamics showed a greater improvement in specific affective and negative symptoms. Moreover, we further indicated that the summative scores of phenotypic behaviors for depression are not well-suited for dissociating the depression subtypes and measuring the treatment outcome. In conclusion, our results suggest that studying whole-brain dynamics could have profound implications for identifying reliable biomarkers and neurostimulation targets for the treatment of psychiatric disorders.

## 1. Introduction

Repetitive transcranial magnetic stimulation (rTMS), which consists of temporally structured magnetic pulse trains, is an emerging therapy for treatment-resistant depression [1]. A large sample meta-study of randomized, sham-controlled trials has provided evidence supporting the therapeutic value of rTMS in depression [2]. There are several rTMS protocols approved by the U.S. Food and Drug Administration. The conventional rTMS protocols entail either 10 Hz rTMS delivered to the left dorsolateral prefrontal cortex (dlPFC) or 1 Hz rTMS over the right dlPFC. The mechanism of action is still largely unknown, but it has been suggested that it involves modulation of the local excitability of the neurons in the frontal cortex, which in turn causes plasticity and leads to modulation of the neural signaling within subcortical networks [3].

While rTMS is an effective treatment for decreasing the overall symptoms of depression, it has been found that conventional rTMS protocols over the dlPFC may not significantly treat symptoms related to anhedonia and avolition (i.e., loss of pleasure and motivation) in patients with depression [4,5]. Specifically, anhedonia and avolition symptoms have been linked to the reward system and dorsomedial prefrontal cortex (dmPFC) dysfunction [6–8]. Moreover, neuromodulation of medial prefrontal structures in rodent models has been found to have an effect on striatal dopaminergic signaling, giving rise to changes in reward-seeking behavior [9]. The dmPFC is thought to act as the convergence site for the cognitive control and affect-triggering networks, and thereby it plays a critical role in both emotion generation and regulation [10]. Accordingly, the dmPFC has recently gained attention as a potential beneficial target for rTMS treatment in depression [1,11], although this has been challenged in a recent placebo-controlled trial of high-frequency rTMS [12]. Nevertheless, our group recently provided evidence supporting the effect of targeting dmPFC with a more intensive stimulation protocol, an accelerated intermittent theta-burst stimulation (iTBS), for the treatment of anhedonia, avolition, and blunted affect in patients with depression [11]. The iTBS protocol significantly reduces the session duration while being non-inferior to the previously established protocols

[13]. Moreover, there is some evidence that multiple daily iTBS sessions (known as accelerated protocols), compared with the single-daily sessions, not only reduce the treatment duration even more, but also improve the observed neuroplasticity effect [14].

That being said, the existing rTMS protocols show high inter-individual variability in the treatment outcome and efficacy, and it is not yet understood how we might match different TMS protocols to the patients [15,16]. The current practice involves trial and error until finding an optimal protocol (in an almost infinite parameter space). Accordingly, the time and cost of investigating alternative stimulation protocols for each person hamper the development of tailored therapies. We argue that extensive whole-brain computational modeling alongside multimodal neuroimaging can help us better characterize depression and facilitate stratified therapy design [17,18]. Relatedly, whole-brain modeling has been found to be a useful tool for finding the hidden dichotomy in the neural dynamics of a seemingly coherent cohort, like attention deficit hyperactivity disorder, with a great potential for guiding stratified neurostimulation therapies [18,19]. Importantly, nonlinear dynamics-based modeling unifies and maps multiple biophysical and behavioral phenomena onto a central level of convergence, with profound implications for how we diagnose and stratify psychiatric disorders and, subsequently, how we decide about the type and target of interventions [18–20]. Using this strategy to stratify patients into treatment groups with customized stimulation protocols could be a major leap towards precision medicine.

In the present study, we evaluated the predictive value of whole-brain modeling for the therapeutic effect of applying accelerated iTBS over dmPFC for patients with depression, using a brief resting-state scan at baseline. We explored our depression sample using computational modeling to unveil covert subtypes exhibiting different whole-brain dynamics. Further, we aimed at testing if the detected subtypes differed in response to rTMS and symptom characteristics. Finding such a dichotomy further stresses the importance of stratification into treatment groups as a key step towards precision medicine.

## 2. Materials and methods

### 2.1 Ethics statement

The study involved human participants and was accordingly reviewed and approved by the Central Ethical Review Board, Uppsala University, Sweden. Moreover, all aspects of the study comply with the ethical standards of relevant human experimentation and with the Helsinki Declaration of 1975, as revised in 2008. Written informed consent was obtained from the research participants.

### 2.2 Study design and participants

This study uses data from a randomized parallel double-blind sham-controlled trial that was pre-registered at the U.S. National Library of Medicine (i.e., ClinicalTrials.gov), with the registration ID of NCT02905604. In the current work, the sample size was based on a previous study with a similar design, in which the response rate was indicated as 17% for sham treatment and 52% for active treatment in the depression cohort [11,21]. Hence, considering a responsive rate of 20% for sham and 52% for active treatment, a sample size of 41 was found for the statistical test with 99% power and 0.05 as the critical value for significant detection (i.e., alpha).

The 42 age-eligible (18–59 years old) patients (mean age: 29 ± 9.4, 21 women) who were included in the current study were recruited from the psychiatric outpatient clinic at the Uppsala University Hospital, Sweden. This data collection started in September 2016 and ended in January 2020. The experiment came to an end after collecting the required amount of data

from the intended number of patients, as determined by the sample size assessment. All patients were diagnosed with either uni- or bipolar depression (n = 3) and fulfilled the criteria of an ongoing depressive episode determined through the Mini International Neuropsychiatric Interview [22]. Moreover, all patients scored at least 40 points on the Motivation and Pleasure Scale—Self Report and had various ongoing but unchanged antidepressive pharmacotherapy over the past month. Patients with pregnancy, epilepsy, implanted devices (e.g., pacemakers), or active substance use disorder (except for nicotine and caffeine) and benzodiazepine use were excluded. An rTMS safety screening questionnaire and the Drug Use Disorders Identification Test were used to identify the exclusion criteria. Patients were randomized to receive either active or sham accelerated iTBS treatment. Moreover, a resting-state fMRI was performed before the treatment.

## 2.3 Treatment

We applied the neuro-navigated iTBS [23] to dmPFC with the Montreal Neurological Institution coordinates of x = 0, y = 30, z = 30 [11,24], using the stimulator MagPro X100 and the MagVenture coil (Cool-DB80 A/P) which is a combined active/placebo coil with two identical sides. It has previously been reported that applying rTMS over the medial frontal cortex, using the same coil type increases the cerebral blood flow in the dmPFC [25]. Individual coil placement was determined using an MRI-based neuronavigation system (TMS Navigator, Localite, Bonn, Germany) in order to precisely target the dmPFC, according to each individual's anatomical T1-weighted scan. The targeted dmPFC was situated deep within the brain, and therefore we defined the strength of stimulation based on the motor threshold of the foot instead of the hand [26]. The corresponding motor cortex lies at a similar depth as our target region, which ensured that our stimulation intensity was strong enough to reach the dmPFC [27]. We estimated the resting motor threshold by applying single pulse TMS over the medial primary motor cortex of the extensor hallucis longus [28], and measuring the sufficient intensity required to elicit an observable muscle contraction in the foot for 50% of the trials, via an automated maximum likelihood strategy [29]. Initially, we gradually ramped up the intensity to maximize comfort and allow the patients to become acquainted with the sensation.

The iTBS treatment was applied twice daily with a 15-min interval to promote plasticity [26,30]. Each session consisted of 40 stimulation trains, all containing two seconds of stimulation, and eight seconds off. Every stimulation interval comprised of 10 bursts at 5 Hz, where three biphasic 50 Hz pulses were delivered in each burst. To achieve bilateral dmPFC stimulation [28], halfway through each session, we reversed the direction of the current (from left-to-right to right-to-left). Reversing the direction of current has been previously shown to induce contralateral motor response in lower extremities, when applied over the motor cortex [27]. The stimulation was applied at 90% of the resting motor threshold [28]. We aimed for 10 treatment days to reach the predetermined target intensity. Yet, if a treatment day did not meet these criteria, the treatment course was prolonged, with a maximum of 15 days [26]. Therefore, any potential low intensity treatment was compensated with the extra days providing comparable effective dosage across individuals. An identical stimulation protocol was applied for the sham treatment with the shield side of the coil pointed towards the patient. Our chosen coil is capable of delivering deep stimulation and is specially designed for sham-controlled studies. The coil consists of two 120° angled figure-eight-shaped coils that are jointed back-to-back. The sham side of the coil is shielded insofar as 95% of magnetic pulses are prevented from reaching the brain [11,26].

Moreover, in all patients, two transcutaneous electrical nerve stimulation (TENS) electrodes were placed on the forehead directly beneath the center of the coil. Thereby, a current of up to

4 mA, synchronized with the TMS pulses and scaled to the magnetic stimulator's output intensity, was delivered in the sham group, to simulate the tactile sensation of active stimulation on the scalp. Therefore, similar to the active treatment, sham treatment patients might require extra treatment days to achieve the intended strength of the TENS stimulation. Notably, because of this aspect of the study, we were able to have a similar number of treatment days for the active and sham groups.

Prior to the first treatment session, the type of treatment (i.e., active or sham) was decided through a randomization process (computerized pseudorandom number with Mersenne Twister algorithm), determining which side of the coil (i.e., not shielded or shielded) to use for each patient. Specifically, a randomization code was prepared by a third person other than the magnetic stimulator's operating nurse and entered in the stimulator's research software. Consequently, the software guided the operator as to which side of the coil should be angled towards the patient. Not to mention that if the patient was assigned to the sham treatment group, the research software automatically activated the TENS stimulation. Moreover, during treatment, daily monitoring for potential adverse effects was conducted. No serious harm or unintended effect was observed in either of the treatment groups. It is worth to further noting that there was no difference in the number of treatment days across active and sham groups [11]. Additionally, to assess the efficacy of our blinding strategy, patients were asked to predict the treatment allocation on the first, fifth, and final treatment days.

## 2.4 Phenotype characterization and behavioral measurements

All participants were assessed using different symptom ratings and executive functioning tests at baseline and after the treatment, including Montgomery Åsberg Depression Rating Scale, self-report (MADRS-S), Brief Psychiatric Rating Scale (BPRS), Clinical Assessment Interview for Negative Symptoms (CAINS) and Trail Making Test (TMT).

**2.4.1 Montgomery Åsberg Depression Rating Scale, Self-report (MADRS-S).** MADRS-S is a self-reported questionnaire to assess the depressive symptoms in patients with mood disorders, based upon the individual's ratings in the range of 0 (none) to 6 (severe) [31]. A summary measure of all scores including {1} mood, {2} feelings of unease, {3} sleep, {4} appetite, {5} ability to concentrate, {6} initiative, {7} emotional involvement, {8} pessimism, {9} suicidal ideation (zest for life) was used to assess the severity of symptoms (referred to as MADRS-S total).

**2.4.2 Brief Psychiatric Rating Scale (BPRS).** The BPRS is a widely used test based on a clinician's interview with the patient, that is used to comprehensively measure psychiatric symptoms ranging from affective to psychotic symptoms. We used the extended 24-item version [32]. The rating scale consists of anchored items ranging from 0 (none) to 7 (severe), utilizing both patient report and observations during the interview to inform the rating. In the current study, we only used the affective subscale of the BPRS (BPRS AFF) to capture clinician-rated affective symptoms [33] based on the summation of scores in depression, suicidality, and guilt measures [33]. Hence, the five included items were: {1} somatic concerns, {2} anxiety, {3} depression, {4} suicidality and {5} guilt.

**2.4.3 Clinical Assessment Interview for Negative Symptoms (CAINS).** The CAINS is a semi-structured interview, mainly performed by RB in this study, and a rating scale with items ranging from 0 (none) to 4 (severe), assessing motivation and pleasure (based on patient report), and emotional expression (based on observation of the patient during the interview) [34]. The motivation and pleasure subscale consists of the following nine items: {1} motivation for close family/spouse/partner relationships, {2} motivation for close friendships and romantic relationships, {3} frequency of pleasurable social activities, {4} frequency of expected

pleasurable social activities, {5} motivation for work and school activities, {6} frequency of expected pleasurable work and school activities, {7} motivation for recreational activities, {8} frequency of pleasurable recreational activities, {9} frequency of expected pleasurable recreational activities. The emotional expression subscale comprises the following four items: {10} facial expression, {11} vocal expression, {12} expressive gestures, and {13} quantity of speech. Accordingly, the primary outcomes include a total score (CAINS total; summation of scores on all items), the motivation and pleasure subscale (CAINS-MAP, summation of items 1–9), as well as the expression subscale (CAINS-EXP, summation of items 10–13).

**2.4.4 Trail Making Test (TMT).** The patients completed a computerized cognitive test battery starting with the TMT, which is a test of visual attention and task switching. TMT consists of two tasks with an increasing level of difficulty from task A to task B [35]. In task A, the circles are numbered 1–25, and the subject is asked to connect them in an ascending order, as quickly as possible. In task B, the circles are labeled with either a number (1–13) or a letter (A–L), and the person is asked to draw lines connecting them in an ascending pattern, with the task being alternated between the numbers and letters (i.e., 1-A-2-B-3-C, etc.). Accordingly, the time to complete a trail making task in seconds (TMT A and TMT B) is used as a measure of executive function. We included TMT as a proxy for executive function in the exploratory analysis as this was the first test in the battery and thus not affected by previously administered tests.

After the last day of active (i.e., iTBS over dmPFC) or sham treatment, all participants completed the same tests as they did prior to treatment. High scores in all the above-mentioned tests reflect more severe symptoms. The main objective was a treatment-related reduction in different depressive symptoms from baseline to after the rTMS treatment course.

## 2.5 Acquisition and prepossessing of resting state functional MRI data

We acquired T1-weighted and functional scans in a Philips 3T Achieva scanner (Philips Medical Systems, Best, Netherlands), with a 32-channel head coil. The image acquisition consisted of structural T1-weighted scans with a 3D Turbo Spin Echo sequence and the following parameters: TR/TE = 8.2/3.8 ms, flip angle = 8˚, field of view = $256 \times 256$ mm$^2$, voxel size = $1 \times 1 \times 1$ mm$^3$ isotropic voxels, 220 slices; and 7 minutes long eye open resting state fMRI (rs-fMRI) obtained using single-shot gradient echo-planar imaging with interleaved acquisition with parameters: 32 slices, TR/TE = 2000/30 ms, field of view = $192 \times 192$ mm$^2$, voxel size = $3 \times 3 \times 3$ mm$^3$ isotropic voxels.

We performed standard fMRI preprocessing and denoising steps, using Statistical Parametric Mapping 12 (SPM12) and functional connectivity toolbox (CONN 20.b) in MATLAB 2021b. To summarize the preprocessing steps, we conducted functional realignment and unwarping, artifact detection by artifact detection tool (using an intermediate setting with 97%-percentiles), segmentation and direct normalization of both functional and structural data to the MNI space, and finally spatial smoothing of functional data with a Gaussian kernel with a size of 6 mm full width at half maximum.

We further accomplished the denoising by applying a linear regression of confounding effects, including nuisance signals from white matter and cerebrospinal fluid areas [36], as well as the twelve motion parameters and their first-order derivatives, the identified outlier scans, and also rest and linear trend effects (to reduce the influence of initial magnetization transients and slow trends). Finally, we applied band-pass filtering [0.008, 0.1 Hz] to the blood-oxygen-level-dependent signals (BOLD) signals to focus on slow-frequency fluctuations while minimizing the influence of non-physiological, head-motion and other noise sources. Filtering was implemented by applying *conn_filter.m* function, implemented in Conn toolbox [37], using a

discrete cosine transform windowing operation to minimize border effects. The length of the window for computing the discrete cosine transform of time-series was set to the length of each time-series.

Moreover, we assessed frame-wise displacement (FD) [38], where we found $FD = 0.25 \pm 0.14$, indicating that our sample falls within an acceptable range. Notably, considering that the scans with an excessive movement were detected and controlled in the denoising step, estimated *FD* was improved to $0.22 \pm 0.09$.

## 2.6 Whole-brain computational modeling of resting-state network

To be able to assess the resting-state functional architecture of the brain beyond the conventional large sample statistical approach, we used our previously developed whole-brain model that is based on weakly coupled oscillators [18,20]. Specifically, the angular frequency dynamics, in addition to the amplitude dynamics, has been incorporated in our model, which has been found to improve its performance [20].

Here, every individual's brain was parceled into 68 regions based on Desikan-Killiany parcellation, and each region was modeled as a nonlinear oscillator. To achieve the macroscopic computational model of the brain, local oscillators are coupled through a structural connectome which was obtained by applying streamline tractography to the large sample diffusion-weighted MRI data from HCP 500 release [39,40]. The structural connectivity matrix (i.e., C) was created by averaging the individual-level streamline densities with a consistency above 60%, followed by resampling the data to follow a Gaussian distribution with $\mu = 0.5$ and $\sigma = 0.15$ [18,20,41].

When coupled together, the collective behavior of interacting oscillators reproduces key features of brain dynamics, including functional connectivity (FC) patterns. The dynamics of each oscillator is defined by coupled differential equations as below:

$$\begin{cases} \dot{z}_j = (A + i\omega_j - |z_j|^2)z_j + G\sum C_{ij}(z_i - z_j) + \beta\eta_j \\ \dot{\omega}_j = \omega^0_{\ j} - F\omega_j + M\sum C_{ij}arg(z_i) \end{cases} \tag{1}$$

Where, z and ω are each oscillator's state variables and describe the regional amplitude and angular frequency. The real part of the variable z (i.e., Rel{z}) acts as an indirect measure of brain activity acquired by the MR scanner, whereas the imaginary part (i.e., Imag{z}) can be interpreted as the hidden state of the system that is unobservable. In the Eq 1, Cij represents the level of structural connectivity between each pair of regions, which was global-wise normalized so that the structural connectivity weights of each region summed up to 1. The $\omega_j$ variable represents the freely running angular frequency of region *j*, which was modeled by a first order differential equation, defined in the second line of Eq 1. $\omega^\circ_{\ j}$ denotes the intrinsic angular frequency for each region which was experimentally obtained [20]. We estimated $\omega^\circ_{\ j}$ at the individual level by finding the peak of the BOLD spectrum. Subsequently, the median value across the cohort was used as the group-representative value for $\omega^\circ_{\ j}$ in the range of 0.098– 0.221 rad/sec, **S1 Fig** Hence, the intrinsic angular frequencies represent the region-specific dominant oscillations during resting state, **S2 Fig** Moreover, the model has four control parameters including A (global bifurcation parameter), G (global coupling parameter), F (global feedback coefficient) and M (global modulation index) that are all scalar values controlling the overall dynamics of coupled oscillators. G and M serve as tuning parameters to scale the structural connectome and to ensure that the local oscillators are weakly coupled together. Plus, an additive Gaussian noise (denoted as ηj, with the standard deviation of β) was implemented as Wiener process to simulate random processes in brain [20]. In order for our search

parameters to remain in the same range with that of the previous research [18,42], we fixed the noise standard deviation to $\beta = 0.02$.

## 2.7 Parameter optimization

In order to fit the model to the fMRI data, we need to find an optimal working point, which is a combination of parameters (i.e., A, G, F and M) within the search interval that maximizes the similarity between the simulated and empirical FC patterns and captures the intrinsic attractor dynamics of the brain. Therefore, the similarity between the empirical and simulated FC was measured to quantify the goodness of fitting. Accordingly, in the grid-search framework, we searched for parameter A from −0.2 to 0.2 with a step-size of 0.01, G from 0.005 to 0.05 with a step-size of 0.005, F from 0 to 1 with a step-size of 0.1, and M from 0.10 to 0.50 with a step-size of 0.03.

Our search protocol resulted in 63,140 simulated sets of whole-brain signals. It is worth noting that the real part of state variable z was taken as the simulated regional BOLD signal. Next, simulated BOLD signals were band-pass filtered (0.008–0.1Hz), followed by computing and the Pearson correlation coefficients among all regions. Identical analysis was performed on the empirical data to achieve the empirical FC matrices. Of note, Fisher's z-transformation was applied to both simulated and empirical FC measures with the aim of performing a subsequent correlation between the simulated and empirical FCs as an index of the goodness of fit of the model. The FC maps of a varying percentage of the patients (i.e., Monte Carlo thresholds) were randomly selected over 1500 Monte Carlo iterations to find the optimal parameter-sets within the search intervals for the whole-brain model. We sampled the FC maps for 500 iterations per threshold (i.e., 30%, 50% and 70%) to remove the dependency of our results on the percentage of patients randomly selected in Monte Carlo resampling procedure. We assessed the similarity between the simulated and empirical FC by taking the correlation between the upper triangles of the simulated and empirical FC matrices in each iteration of Monte Carlo resampling [18]. We found bimodal distributions for all three Monte Carlo thresholds (**S3 Fig**), and therefore we pooled all three distributions to yield a multi-threshold distribution with 1500 resampling. Applying the probabilistic nonparametric framework (i.e., Monte Carlo permutation) enabled us to find the two hidden subtypes (i.e., DEP1 and DEP2) in our depression cohort, which corresponded to the two different peaks in the parameters' distribution. For more details we refer the reader to Iravani *et al* [18].

Next, based on the similarity between each individual's empirical FC and either of the two model-derived FC maps (i.e., simulated $FC_{DEP1}$ and simulated $FC_{DEP2}$), we assigned each subject to one of the models (i.e., each model associated with one of the optimal parameter-sets identified within the search intervals). As a control analysis, we used Pearson chi-squared and mixed-effect models to ensure that the stratification was not related to other factors like treatment allocation and motion.

## 2.8 Statistical analysis

All statistical analysis was performed using Statistical and Machine Learning toolboxes in MATLAB 2021b. At the optimal working point within the search interval, our whole-brain model simulated BOLD signals with FC patterns similar to the empirical resting state data. Based on the similarity between each individual's empirical and simulated FC, we assigned patients to one of the identified depression models/subtypes.

We further indicated that there is no evidence that these two subtypes are allocated to treatment groups in a biased manner using a Pearson's chi squared test. Yet to remove any potential bias driven by having different numbers of individuals associated with subtypes across two

treatment groups, we used bootstrap method for hypothesis testing. Thereby, we equalized the number of individuals assigned to each subtype in every treatment group using resampling with replacement as implemented in the bootstrapping method.

At baseline, we fitted separate logistic regression models (LoM) for MADRS-S total, BPRS AFF and CAINS. Hence, in total, three LoMs were fitted alongside with 1000-bootstrap resampling to assess the effect of depression subtype on the individual items of the three depression symptom ratings (i.e., MADRS-S total, BPRS AFF and CAINS), S4 Fig.

In the follow-up session, subsequent to the treatment (i.e., either active or sham iTBS over dmPFC), the behavioral measures consisting of MADRS-S total, BPRS AFF, CAINS and TMT were collected and contrasted to values at the baseline to generate the change score across two sessions. The main effects of treatment (active or sham), depression subtypes and their interaction on behavioral change scores (follow-up session contrasted to the baseline session) were assessed using a linear regression model (LiM) and 1000-bootstrap resampling of the coefficients (S5 Fig). We accounted for the baseline behavior measures, age and gender in our LiMs, by including them in the fixed-effect term.

In the follow-up analysis, we used a nested ANOVA model to determine if there was any difference between the depression subtypes nested in the treatment, explaining the change in the individual symptom ratings. Age and sex were controlled in the nested ANOVA model. We complemented our analysis with a subsequent post-hoc 1000-resampling bootstrap test to determine which depression subtype showed more improvement after receiving the active treatment, with regard to individual items of MADRS-S, BPRS AFF and CAINS. It should be further noted that the change score of each individual item was quantified as symptom improvement for active treatment contrasted against the median improvement of sham treatment.

## 3. Results

### 3.1 Baseline data

There were no statistical differences in demographics or baseline summative clinical measures across the two treatment groups, as well as the two detected depression subtypes Table 1.

**Table 1. Demographics and baseline clinical data.**

|  | Active treatment (iTBS) | Sham treatment | Model based delineation, DEP1 | Model based delineation, DEP2 |
|---|---|---|---|---|
| Sex, N (male/female) | 12/10 | 9/11 | 12/8 | 9/13 |
| Age, mean (SD) | 30 (10) | 29 (9) | 32 (10) | 27 (8) |
| Handedness (left/right) | 4/18 | 2/18 | 5/15 | 1/21 |
| Baseline symptoms, mean (SD) |  |  |  |  |
| MADRS-S total[a] | 30 (8) | 29 (7) | 30 (9) | 29(6) |
| CAINS total[b] | 28 (8) | 30 (8) | 29 (7) | 29 (8) |
| CAINS MAP[c] | 22 (6) | 21 (6) | 21 (5) | 21 (7) |
| BPRS[d] | 2.0 (0.4) | 2.1 (0.2) | 2.1 (0.4) | 2.0 (0.2) |
| TMT A[e] | 1.7 (0.2) | 1.8 (0.2) | 1.8 (0.2) | 1.7 (0.2) |
| TMT B[f] | 1.9 (0.2) | 1.8 (0.2) | 1.9 (0.2) | 1.8 (0.2) |

[a] Montgomery Åsberg Depression Rating Scale, Self-report, total.

[b] Clinical Assessment Interview for Negative Symptoms.

[c] Clinical Assessment Interview for Negative Symptoms–motivation and pleasure subscale

[d] Brief Psychiatric Rating Scale

[e] Trail Making Test–task A

[f] Trail Making Test–task B

## 3.2 Whole-brain dynamics of depression is heterogeneous

Parameter-sets were estimated by non-parametric fitting of the model to the resting-state FC at baseline (**Fig 1A**), indicating a bimodal distribution (**Fig 1B**). The two peaks capture different attractor dynamics, representing hidden subtypes in our depression cohort, here called

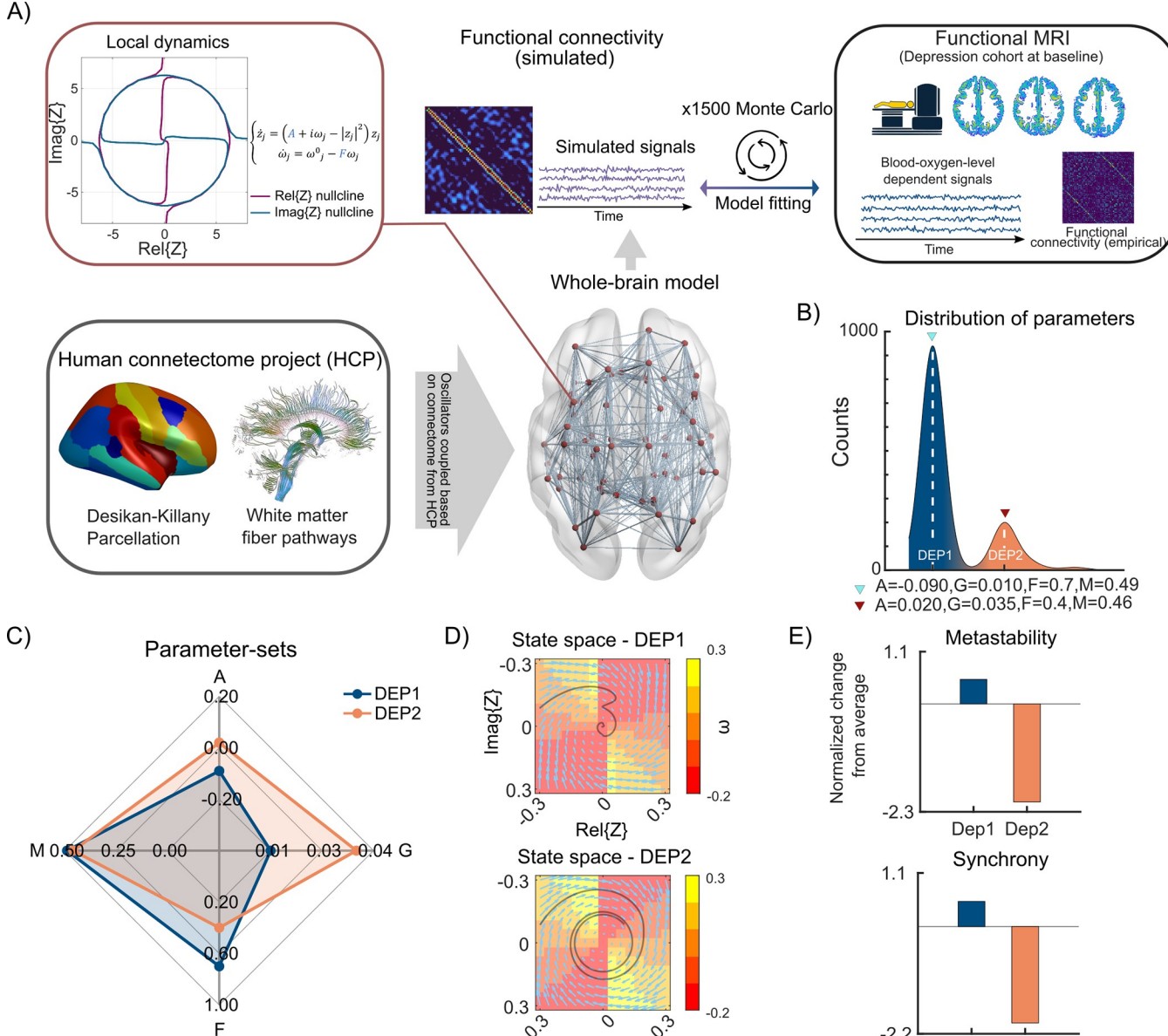

**Fig 1. Whole-brain modeling stratifies the depression sample.** (A) The whole-brain model was constructed from oscillators with amplitude and angular frequency dynamics, which were coupled through the structural connectome. The model parameters were optimized in an iterative Monte Carlo procedure based on the similarity between model-derived functional connectivity with that of the empirical data. The white matter fiber tracks figure was adopted from Wikipedia.com (https://en.wikipedia.org/wiki/Diffusion_MRI#Diffusion_tensor_imaging) under a CC BY 4.0 license. (B) The distribution of parameter-sets obtained from 1500 Monte Carlo resamplings across three thresholds (i.e., 30%, 50% and 70%) indicated two peaks, representing two subtypes in the data, so-called DEP1 and DEP2. The graphs depict (C) different optimal parameter-sets capturing differences in amplitude and angular frequency dynamics in the depression cohort and (D) their respective state spaces. The black spirals represent the underlying stable focus attractors, while the angular frequency state variable ω is color-coded. (E) The bar graphs denote the normalized difference in metastability and synchrony across the two depression subtypes. DEP1 exhibited higher metastability and synchrony, whereas DEP2 demonstrated lower metastability and synchrony compared to the average of the Monte Carlo resampling distribution.

DEP1 and DEP2 (**Fig 1B and 1C**). The individual-level model predictions can be found in **S1 Table**.

Global feedback parameter F and global modulation index M were larger for DEP1 compared with DEP2; however, the bifurcation parameter A and global coupling parameter G were larger in DEP2 subtype compared to DEP1. A larger bifurcation parameter A indicates a potential higher neural excitability. In a system with a larger global modulation index M, the angular frequency dynamics of individual oscillators (i.e., regions) have higher sensitivity to the subsiding ensemble activity in their immediate neighboring regions (i.e., the ones which are structurally connected together). The attractor space for the two estimated optimal parameter-sets within the search interval was illustrated in **Fig 1D**. It is worth mentioning that given these two parameter-sets, the model produced regional BOLD-like signals with FC maps resembling the empirical FCs of depression subtypes ($r_{DEP1}$ = 0.339 and $r_{DEP2}$ = 0.333). DEP1 demonstrated a 0.5 standard deviation larger metastability and synchrony, however, DEP2 exhibited 2.1 and 2.0 standard deviations lower metastability and synchrony compared to the mean of the resampled Monte Carlo distribution, **Fig 1E** and **S1 Table**. The simulated time-series associated with DEP1 and DEP2 as well as their empirical counterparts are illustrated in **S6 Fig**. It should be pointed out that the objective of our model was to simulate the FC patterns similar to the FCs of the empirical data rather than simulating the individual time-series. The mean FCs for DEP1 and DEP2 are illustrated in the **S7 Fig**. The similarity between the two subtypes' FC was *rho* = 0.95.

To ensure that the two unraveled subtypes (i.e., DEP1, n = 20 and DEP2, n = 22, for more information on the demographic and treatment allocation of depression subtypes, please refer to **S1 Table**) were not driven by differences in treatment allocation, as a control analysis, we initially investigated whether the number of patients designated to the two subtypes were comparable across active and sham groups. Pearson's Chi-squared test ($\chi^2(1)$ = 0.09, $p > .77$) indicated that there was no association between the depression subtypes and the treatment groups (i.e., there was no significant bias in treatment for depression subtypes). Furthermore, we used a linear mixed-effect model with age, gender, treatment group, depression subtype as well as interaction of depression subtype and treatment group as independent variables with a by-participant random intercept to assess if there was any head motion effect that was driving the observed grouping. There were neither a difference in *FD* as a function of treatment, $t(36)$ = 1.32, $p > .19$, nor depression subtypes, $t(36)$ = -0.55, $p > .59$, nor their interaction $t(36)$ = 1.23, $p > .23$.

We also assessed whether the two subtypes were associated with the comorbidity of depression with anxiety, autism, attention deficit hyperactivity disorder or personality disorder, using Pearson's chi-squared tests. We found no association between the detected subtypes and the comorbidity with any of the above-mentioned disorders (all $ps > .14$).

## 3.3 Individual factors of MADRS-S total, BPRS AFF and CAINS differentiate depression subtypes at baseline

Our modelling approach resulted in the detection of two covert subtypes exhibiting different pre-treatment attractor dynamics. We asked whether the specific symptoms of MADRS-S total or BPRS AFF or CAINS dissociated the depression subtypes at baseline. Therefore, we fitted a LoM, separately for MADRS-S total, BPRS AFF and CAINS. We found that some individual items of MADRS-S total, BPRS AFF and CAINS at baseline were related to the depression subtypes. Particularly, we found larger odds ratio for MADRS-S2 (*Feeling of Unease*), 95% $CI_{bootstrap}$ = [0.22, 0.93] and MADRS-S6 (*Initiative*), 95% $CI_{bootstrap}$ = [0.26, 0.83], in DEP1 compared with DEP2, whereas the odds ratio of MADRS-S4 (*reduced Appetite*), 95%

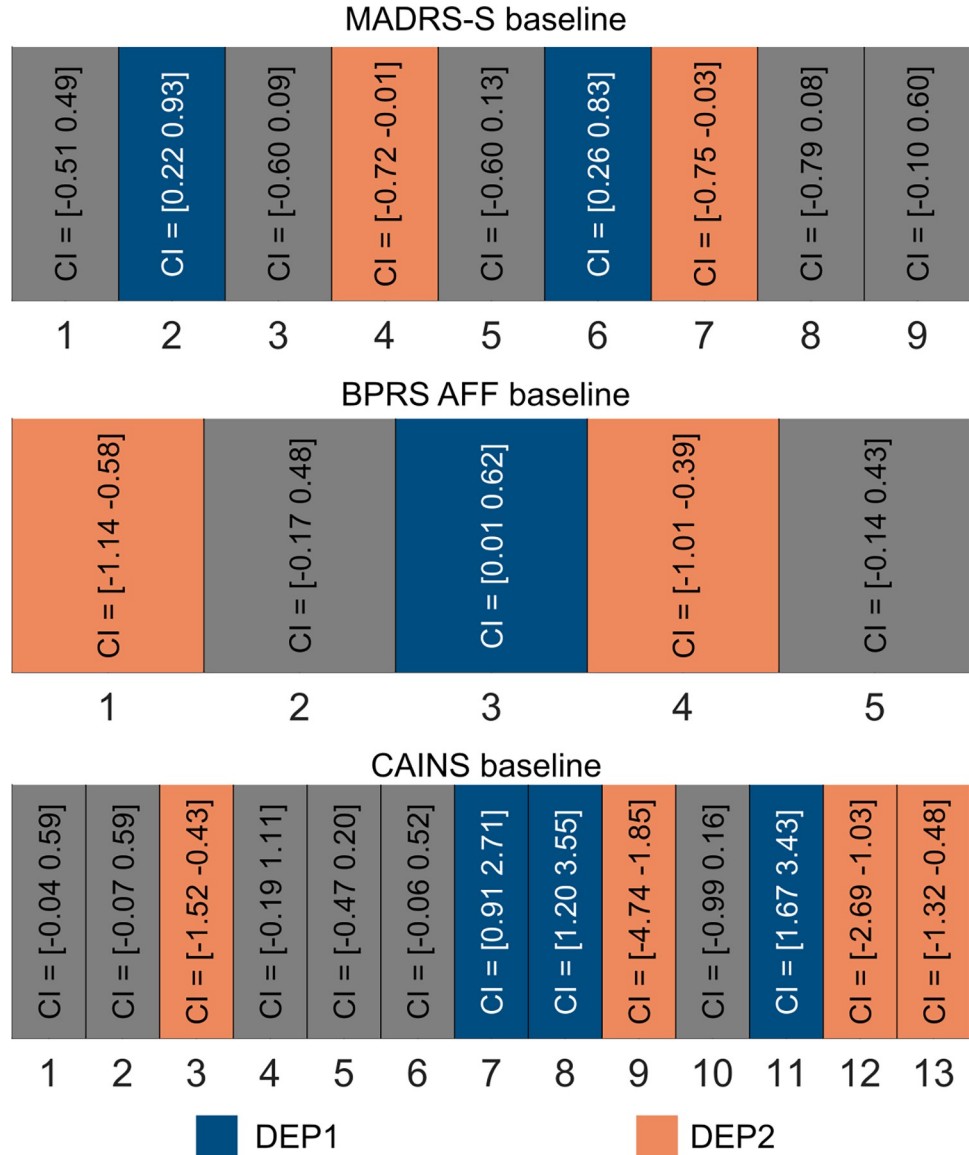

**Fig 2. Specific symptoms of MADRS-S, BPRS AFF and CAINS predict depression subtypes at baseline.** The bootstrap 95% confidence intervals of the logistic regression coefficients for MADRS-S, BPRS AFF and CAINS at baseline. The numbers on the horizontal axes of each panel correspond to the symptoms listed in Treatment section for a given symptom rating. The blue and orange colors indicate the individual items that were significantly different for a specific depression subtype.

$CI_{bootstrap}$ = [0.01, 0.72], as well as MADRS-S7 (*Emotional Involvement*), 95% $CI_{bootstrap}$ = [0.03, 0.75], were larger in DEP2 compared with DEP1 (*Fig 2*, upper row).

We also observed that the odds ratio of BPRS AFF3 (*Depression*), 95% $CI_{bootstrap}$ = [0.01, 0.62], was larger for DEP1 compared to DEP2. However, BPRS AFF1 (*Somatic Concerns*), 95% $CI_{bootstrap}$ = [0.58, 1.14] and BPRS AFF4 (*Suicidality*), 95% $CI_{bootstrap}$ = [0.39, 1.01], were larger in DEP 2 compared to DEP1 (*Fig 2*, middle row).

CAINS7 (*Motivation for Recreational Activities*), 95% $CI_{bootstrap}$ = [0.91, 2.71], CAINS8 (*Frequency of Pleasurable Recreational Activities*), 95% $CI_{bootstrap}$ = [1.20, 3.55] and CAINS11 (*Vocal Expression*), 95% $CI_{bootstrap}$ = [1.67, 3.43], were higher in DEP1 compared to DEP2

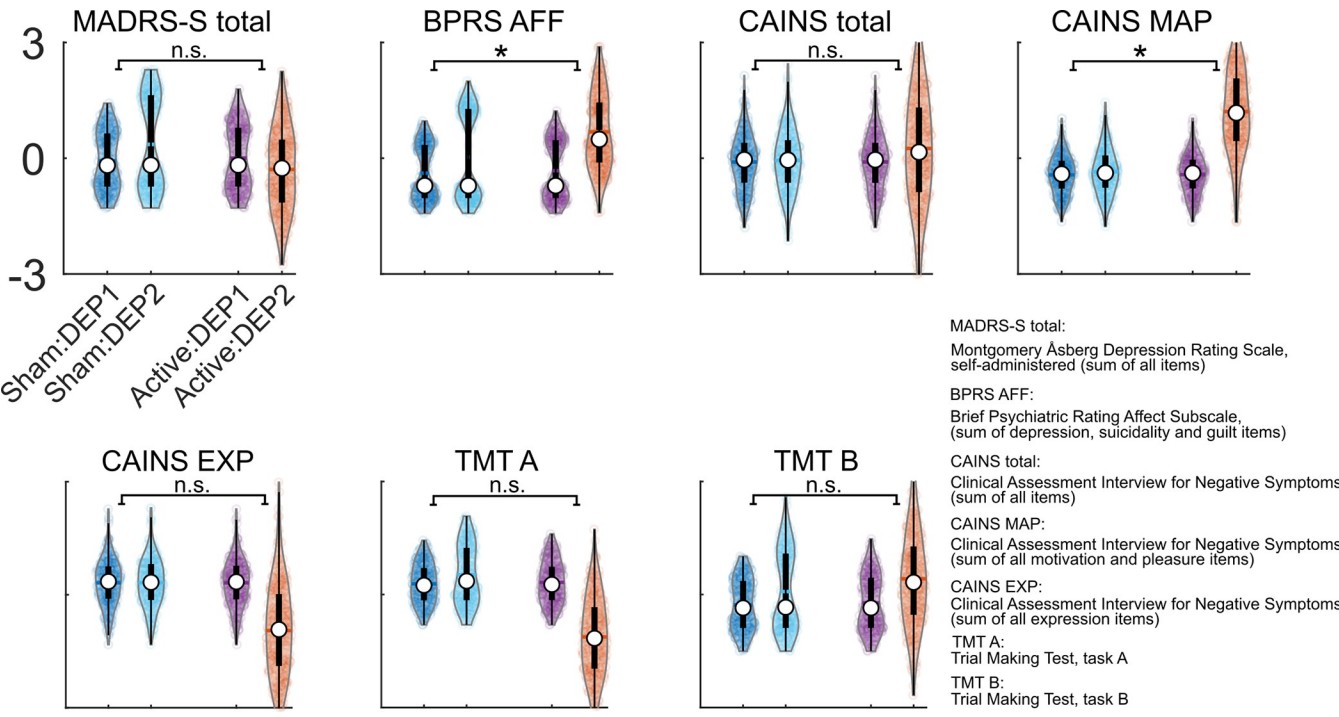

**Fig 3. Depression subtypes modulate the changes in behavior.** Violin plots show the distributions of changes in MADRS-S total, BPRS AFF, CAINS total, CAINS motivation and pleasure subscale (CAINS MAP), CAINS expression subscale (CAINS EXP), TMT A and TMT B across two depression subtypes and treatments. Within each violin plot, there is a box plot summarizing the bootstrap results by indicating the median (depicted by white dot), the first and third quadrants, represented by the lower and upper edges of the black box, and interquartile range by the whiskers. The dots within violin plots represent the resampled data points. Note, the full name of the behavioral measures can be found in the bottom right corner of the figure.

(**Fig 3**). Conversely, CAINS3 (*Frequency of Pleasurable Social Activities—Past Week*), 95% $CI_{bootstrap}$ = [0.43, 1.52], CAINS 9 (*Frequency of Expected Pleasurable Recreational Activities– Next Week*), 95% $CI_{bootstrap}$ = [1.85, 4.74], CAINS12 (*Expressive Gestures*), 95% $CI_{bootstrap}$ = [1.03, 2.69] and CAINS13 (*Quantity of speech*), 95% $CI_{bootstrap}$ = [0.48, 1.32], had higher odd ratio for DEP2 compared to DEP1 (**Fig 2**, lower row).

The bootstrap distribution for individual items of MADRS-S, BPRS AFF and CAINS at baseline and follow-up can be found in the **S4 Fig**.

## 3.4 The therapeutic effect of iTBS on dmPFC is modulated by intrinsic brain dynamics at baseline

Prior to assessing the effect of iTBS and the depression subtypes, we checked the integrity of our blinding. After the first treatment day, two thirds of patients in both groups correctly anticipated the treatment allocation. Yet, because there was no difference in the number of patients who correctly guessed their treatment across the two groups, we regarded it as an adequate level of blinding. It should be noted that the number of patients who correctly anticipated their treatment allocation decreased to two-fifths in the active group and increased to four-fifths in the sham group by the last treatment day [11].

In the previous sections, we indicated that the attractor dynamics at baseline dissociates the depression cohort into two subtypes (i.e., DEP1 and DEP2), exhibiting distinct phenotypical behaviors. Next, we assessed whether the two subtypes responded differently to the treatment. Hence, the cohort was grouped with four levels (i.e., Sham:DEP1, n = 10; Sham:DEP1, n = 10; Active:DEP1, n = 10; Active:DEP2, n = 12) for the LiM analysis. The outcome measures were

summative scores of depression (i.e., MADRS-S total), affective psychiatric (i.e., BPRS AFF) and negative symptoms (i.e., CAINS). Using bootstrapping of the LiM's coefficients, we found a significant effect for the interaction of treatment and depression subtypes, for BPRS AFF, 95% $CI_{bootstrap}$ = [0.37, 3.09] and CAINS MAP, 95% $CI_{bootstrap}$ = [0.05, 5.21] **Fig 3**. No evidence was found for modulation of cognitive function, TMT A, 95% $CI_{bootstrap}$ = [-0.06, 0.00] and TMT B, 95% $CI_{bootstrap}$ = [-0.03, 0.05]. Moreover, we did not find any effect for MADRS-S total, 95% $CI_{bootstrap}$ = [-4.21, 1.76], CAINS total, 95% $CI_{bootstrap}$ = [-2.99, 4.05], and CAINS EXP 95% $CI_{bootstrap}$ = [-1.83, 0.33]. See the **S5 Fig** for the bootstrap distributions of all the LiM coefficients. Moreover, The FC associated with each treatment group and depression subtype is depicted in **S8 Fig**.

### 3.5 The iTBS treatment modulates individual items in a certain depression subgroup

Earlier we found that BPRS AFF and CAINS MAP were modulated by the interaction of the treatment and depression subtype. To increase the sensitivity of our analysis we further assessed if the treatment modulated any of the individual items of the behavioral measure as a function of depression subtype. We used a nested 2-way ANOVA to assess whether the specific symptoms of MADRS-S total, BPRS AFF and CAINS, were modulated by treatment and depression subtypes.

We found that there was an effect of depression subtype nested within treatment groups for MADRS-S1(*Mood*), $F(2,15)$ = 6.87, $p < .008$; MADRS-S6 (*Initiative*), $F(2,15)$ = 5.99, $p < .012$; BPRS AFF3 (*Depression*), $F(2,15)$ = 5.10, $p < .020$; BPRS AFF4 (*Suicidality*), $F(2,15)$ = 9.49, $p < .002$ and CAINS5 (*Motivation for Work and School Activities*), $F(2,15)$ = 3.69, $p < .049$ (**Fig 4A**). We did not

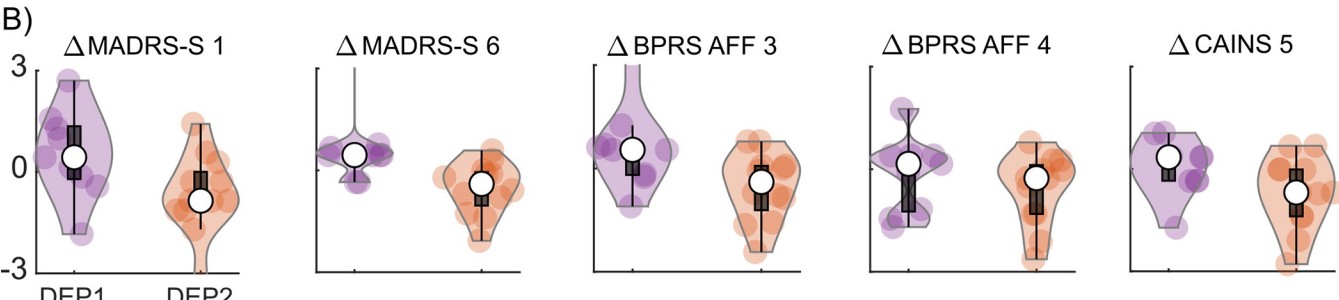

**A)**

**MADRS-S 1**

| Source | Sum Sq. | d.f. | Mean Sq. | F | Prob>F |
|---|---|---|---|---|---|
| age | 40.47443 | 21 | 1.9273538 | 2.59193 | 0.031694 |
| sex | 0.69768 | 1 | 0.6976796 | 0.938248 | 0.348089 |
| treatment(DEP) | 10.2211 | 2 | 5.1105503 | 6.872732 | 0.007611 |
| DEP | 0.245014 | 1 | 0.245014 | 0.329498 | 0.574462 |
| Error | 11.15397 | 15 | 0.7435981 | | |
| Total | 55.01599 | 40 | | | |

**MADRS-S 6**

| Source | Sum Sq. | d.f. | Mean Sq. | F | Prob>F |
|---|---|---|---|---|---|
| age | 27.042365 | 21 | 1.28773166 | 0.974725 | 0.531724 |
| sex | 0.701857 | 1 | 0.70185702 | 0.531258 | 0.477305 |
| treatment(DEP) | 15.835722 | 2 | 7.91786091 | 5.993282 | 0.01222 |
| DEP | 0.0514747 | 1 | 0.05147473 | 0.038963 | 0.846172 |
| Error | 19.816839 | 15 | 1.3211226 | | |
| Total | 54.420432 | 40 | | | |

**BPRS AFF 3**

| Source | Sum Sq. | d.f. | Mean Sq. | F | Prob>F |
|---|---|---|---|---|---|
| age | 41.11714 | 21 | 1.9579591 | 2.258728 | 0.055132 |
| sex | 0.177198 | 1 | 0.1771976 | 0.204418 | 0.657648 |
| treatment(DEP) | 8.842291 | 2 | 4.4211455 | 5.100294 | 0.020422 |
| DEP | 0.012091 | 1 | 0.0120912 | 0.013949 | 0.907552 |
| Error | 13.00262 | 15 | 0.8668413 | | |
| Total | 58.60742 | 40 | | | |

**BPRS AFF 4**

| Source | Sum Sq. | d.f. | Mean Sq. | F | Prob>F |
|---|---|---|---|---|---|
| age | 25.35127 | 21 | 1.2072033 | 2.286274 | 0.052609 |
| sex | 0.029789 | 1 | 0.0297891 | 0.056416 | 0.815466 |
| treatment(DEP) | 10.02297 | 2 | 5.0114858 | 9.491053 | 0.002169 |
| DEP | 0.01453 | 1 | 0.0145298 | 0.027517 | 0.870464 |
| Error | 7.920332 | 15 | 0.5280221 | | |
| Total | 37.55134 | 40 | | | |

**CAINS 5**

| Source | Sum Sq. | d.f. | Mean Sq. | F | Prob>F |
|---|---|---|---|---|---|
| age | 21.81723 | 21 | 1.0389157 | 1.232249 | 0.344164 |
| sex | 0.001408 | 1 | 0.0014076 | 0.00167 | 0.967946 |
| treatment(DEP) | 6.218025 | 2 | 3.1090123 | 3.687574 | 0.049824 |
| DEP | 0.305489 | 1 | 0.3054891 | 0.362338 | 0.556201 |
| Error | 12.64658 | 15 | 0.843105 | | |
| Total | 37.17832 | 40 | | | |

**Legend**

**MADRS-S 1**: Mood                    **BPRS AFF 3**: Depression

**MADRS-S 6**: Initiative                **BPRS AFF 4**: Suicidality

**CAINS 5**: Motivation for work and school activities

**B)**

**Fig 4. Treatment and depression subtypes modulate certain items of MADRS-S, BPRS AFF and CAINS.** (A) The table summarizes the nested ANOVA results for MADRS-S, BPRS AFF and CAINS items. (B) Violin plots depict the distributions of the differences in MADRS-S1, MADRS-S6, BPRS AFF3, BPRS AFF4 and CAINS 5 changes (follow-up–baseline) between active treatment and the median score in the sham treatment for each depression subtype. The boxplot summarizes the statistical parameters of the distribution. The white dot represents the median, the lower and upper edges of the black box show the first and third quadrants, and the interquartile range is shown by the whiskers. The dots within the violin plot represent the individual data points.

find any evidence supporting that subtype nested in treatment modulates other individual items of MADRS-S ($ps > .12$) or BPRS AFF ($ps > .49$) or of CAINS ($ps > .06$). A post-hoc 1000-resampling bootstrap test indicated that the change in the improvements MADRS-S1 (*Mood*), 95% *CI* = [0.58, 1.76]; MADRS-S6 (*Initiative*), 95% *CI* = [0.78, 1.83]; BPRS AFF4 (*Suicidality*), 95% *CI* = [0.27, 1.22] and CAINS5 (*Motivation for Work and School Activities*), 95% *CI* = [0.59, 1.61] was lower in DEP2 compared to the DEP1 subtype (**Fig 4B**). The improvement of BPRS AFF3 (*depression*), 95% *CI* = [-0.06, 0.96], in DEP2 did not survive the post-hoc 1000-resampling bootstrap test.

## 4. Discussion

Whole-brain computational modeling combined with nonparametric probabilistic fitting applied to the pre-treatment resting-state fMRI data from patients with treatment-resistant depression, disclosed a dichotomy in the attractor dynamics of our depression sample. According to the derived optimal model parameters within the search interval, we could stratify the patients into the DEP1 and DEP2 subtypes based on their pretreatment FC patterns. We indicated that the two detected depression subtypes exhibited different phenotypic behavior related to certain affective and negative symptoms at baseline. We further found that the effect of treatment on the summative scores for the affect subscale of BPRS, motivation and pleasure subscale of CAINS varied as a function of depression subtypes. Nevertheless, the effect on the summative scores was mostly driven by the worsening of DEP2 by active treatment, potentially due to the heterogeneity of DEP2's response to individual items of the measurements. We followed up on this observation by assessing the individual items of MADRS-S total, BPRS AFF and CAINS using nested ANOVA. We found that the behavior related to mood, initiative/lassitude, suicidality and motivation for work and school activities was indeed improved in the subtype with a lower metastable and synchronous pretreatment intrinsic brain activity (i.e., DEP2), despite worsening of the summative scores. Hence, our results suggested that the summative scores of phenotypic behaviors for depression are not well-suited for dissociating the depression subtypes.

There are diverging neurophysiological mechanisms involved in depression, and therefore, large-scale multimodal analysis of depression cohorts has failed to uncover distinctive neurobiological biomarkers [43–45]. This lack of consistent findings or surprisingly negligible effects has been partly attributed to the heterogeneity of this clinical population, which in turn limits the improvement of rTMS therapies. Relatedly, the rTMS treatments are associated with high inter-individual variance [15].

Fortunately, there is a growing momentum in psychiatry research to disclose the heterogeneity in the cohorts and unravel different subtypes based on neuroimaging data [18]. For example, Zhang *et al.* [46] identified depression subtypes using a data-driven approach based on static FC analysis, but they did not find any inter-group difference in response to rTMS over the dlPFC [46]. Nevertheless, our approach relies on hypothesis-driven computational modeling to integrate information from structural and functional connectivity and characterize mental disorders at a central layer of convergence linking various neurophysiological and behavioral phenomena. We previously demonstrated that our whole-brain modeling approach is capable of stratifying a seemingly homogenous patient cohort based on their underlying attractor dynamics, which may in turn explain potential differences in the therapeutic effect of neurostimulation [18]. Yet, our previous research lacked the rTMS data to validate this hypothesis in an experimental set-up. This aspect has been fulfilled in the current study. Thereby we were able to evaluate the effectiveness of whole-brain modeling in identifying clinically relevant, yet hidden, subtypes with different responses to the same rTMS therapy, in a sample of patients with treatment-resistant depression. Specifically, the two depression

subtypes exhibited different frequency dynamics. With this regard, we previously showed that impairment in the frequency aspect, especially the stability of instantaneous phase synchrony patterns, is related to impaired cognitive control [47]. Here, the subgroup of patients with higher responsiveness to treatment exhibited blunted metastable dynamics at baseline intrinsic activity, perhaps reflecting a pathological state of frequency lethargy that was possibly alleviated after iTBS therapy. Indeed, the DEP2 subtype had a lower global feedback coefficient F as well as a lower global modulation index M (i.e., weight of neighbor ensemble's net phase, which is inversely related to the neighbor ensemble's activity), indicating blunted frequency dynamics. Whereas observing a higher global feedback coefficient F and modulation index M for the DEP1 subtype, may reflect the highly unstable and variable nature of frequency dynamics (i.e., higher metastabilty) in this subtype. Abnormality of dynamic regional phase synchrony in depression has also been reported elsewhere [48].

It is worth noting we used a new rTMS protocol, namely the accelerated dmPFC iTBS, compared to commonly used non-invasive stimulation protocols for depression (i.e., high- or low-frequency rTMS over the dlPFC). Specifically based on lesion and meta-analysis studies, stimulating the dmPFC has been suggested to be a promising alternative target for neurostimulation therapy in depression [1]. Additionally, the dmPFC is recently referred to as 'dorsal nexus', where the cortical networks of affect regulation, cognitive control and self-reflection exclusively overlap in depression but not in healthy control [1,49]. Critically, the dmPFC has been shown to be linked to anhedonia and avolition symptoms of depression [50].

Nevertheless, applying iTBS over dmPFC has yielded mixed results in treating the overall depressive symptoms [11,12,28]. We argue that the possible reason for the ambiguity in the findings is the heterogeneity in the intrinsic attractor dynamics at the baseline. Relatedly, we found significant improvements in specific symptoms of depression (mood and initiative), suicidality and motivation for work and school activities, in response to iTBS over dmPFC for one of the groups (i.e., DEP2) detected based on baseline attractor dynamics. This highlights the potential of designing stratified treatment protocols based on the brain dynamics of patients at baseline, in order to increase the success rate of rTMS therapies. We speculate that the improved therapeutic success for the DEP2 is due to the effectiveness of iTBS over dmPFC in rectifying the DEP2's specific impaired frequency dynamics. Accordingly, we observed that DEP2 exhibited blunted frequency attractor dynamics (**Fig 1D and 1E**) compared to DEP1, which may explain the remedial leverage of the active treatment in DEP2.

Our study had several limitations, our grid-search spanned a limited interval of the parameter space and therefore the obtained optimal parameter-sets are contingent on our defined search scope. Moreover, the small sample size bounded our model prediction and the statistical analysis. Furthermore, because our nested ANOVA analysis was an exploratory analysis, our findings should be considered as preliminary and should be validated in an independent sample. However, we speculate that with a larger sample, several finer grained depression subtypes might be identified. Another possible limitation of the current study is that instead of using sample-specific structural connectivity, we used the structural connectivity estimated from HCP to minimize the possible errors arising from an under-sampled dataset. Furthermore, we included only 68 cortical regions in our model. Future studies with more realistic computational models based on a higher spatial resolution and larger sample sizes are required to further assess the heterogeneity in depression. Notably, computational modeling based on longitudinal functional data collected during the treatment course as well as the follow-up session are required to shed light on the underlying mechanism of iTBS in depression.

In conclusion, we used a novel approach based on the attractor dynamics of whole-brain model to delineate the depression cohort in a clinically relevant manner. Our approach provided a mathematical platform to efficiently combine the tractography and functional imaging

data, while taking the nonlinear dynamics of the brain into account. We found improvements in mood, initiative/lassitude, suicidality, motivation for work and schools in response to applying iTBS to dmPFC, in the subgroup with blunted frequency dynamics and less global metastability and synchrony. This finding highlights the importance of designing stratified treatment protocols in order to increase the success rate of rTMS therapies. Nevertheless, it should be noted that this type of stratification of the depression cohort is novel and has to be further validated in independent samples before finding its way into clinical practices.

## Supporting information

**S1 Fig. The distribution of intrinsic angular frequencies.** The peak of spectral density that was derived from Welch method was identified for each region at the individual level. The median value across the cohort was used as the group-representative value of $\omega°_j$ in Eq 1. (TIF)

**S2 Fig. Estimated intrinsic angular frequency map.** The intrinsic angular frequency is represented on the Desikan-Killany atlas using a color-coded scheme. Warmer colors indicate a higher intrinsic angular frequency whereas the cooler colors indicate a lower intrinsic angular frequency. (TIF)

**S3 Fig. Distribution of parameter-sets as a function of Monte Carlo threshold.** In Monte Carlo resampling, we randomly selected 30%, 50% or 70% (i.e., Monte Carlo threshold) of patients with 500 iterations per threshold. In all three threshold choices, we observed bimodal distributions with similar peak values (optimal parameter-sets), indicating that two subtypes were detectable at all three thresholds. (TIF)

**S4 Fig. Bootstrap distribution for the logistic regression models.** The distributions in each column represent a coefficient in the logistic regression model for MADRS-S (upper panel), BPRS AFF (middle panel) and CAINS (bottom panel) at baseline. The dashed red line depicts 0, which represents no association with the stratification of the depression cohort. At baseline, the bootstrap 95% confidence interval for MADRS-S2, MADRS-S4, MADRS-S6, MADRS-S7, BPRS AFF1, BPRS AFF3, BPRS AFF4, CAINS3, CAINS7, CAINS8, CAINS9, CAINS11, CAINS12 and CAINS13 did not include 0 at baseline, indicating a statistically significant association with the stratification of the depression cohort. (TIF)

**S5 Fig. Bootstrap distribution for the linear regression models predicting behavioural change from baseline to follow-up session.** The distributions in each column represent a coefficient in the linear regression model where each row corresponds to a behavioral measure (i.e., outcome). The dashed red line depicts 0, which represents no association with the outcome. In this analysis, the coefficient of interest was the last column (treatment_active:subtype (DEP1/DEP2)). For BPRS AFF and CAINS MAP, the bootstrap 95% confidence interval for the treatment_active:subtype(DEP1/DEP2) did not include 0, indicating a statistically significant association with the outcome from baseline to follow-up. (TIF)

**S6 Fig. The simulated and empirical BOLD for major brain lobes. A**) The simulated time-series for the two parameter-sets (i.e., DEP1: A = -0.090, G = 0.010, F = 0.7, M = 0.49 and DEP2: A = 0.020, G = 0.035, F = 0.4, M = 0.46) were averaged for all regions within each major brain lobe (i.e., Frontal, Temporal, Parietal and Occipital). **B**) The empirical BOLD time-series

for each major brain lobe were averaged across individuals within each identified subtype.
(TIF)

**S7 Fig. Functional connectome for depression subtypes.** The empirical functional connectivity (FC) was derived from band-pass filtered BOLD signals. The resulting FCs were Fisher's z-transformed, and any negative values were replaced with 0.
(TIF)

**S8 Fig. Functional connectome as a function of subtypes and treatment.** The band-pass filtered BOLD time-series were used to compute functional connectivity (FC) for depression subtypes and treatment groups. The resulting FCs were Fisher's z-transformed, and any negative values were replaced with 0. The number of individuals contributing to each FC plot is written in the second line of the title.
(TIF)

**S1 Table. Demographic, treatment allocation and dynamical brain measures: individual model-driven prediction and correlations with empirical data.**
(XLSX)

## Acknowledgments

We appreciate the Human Connectome Project for providing the open access data used in this study.

## Author Contributions

**Conceptualization:** Neda Kaboodvand, Behzad Iravani.

**Data curation:** Behzad Iravani, Jonas Persson.

**Formal analysis:** Neda Kaboodvand, Behzad Iravani.

**Funding acquisition:** Neda Kaboodvand, Behzad Iravani.

**Investigation:** Neda Kaboodvand, Behzad Iravani, Jonas Persson, Robert Boden.

**Methodology:** Neda Kaboodvand, Behzad Iravani.

**Project administration:** Martijn P. van den Heuvel, Jonas Persson, Robert Boden.

**Resources:** Martijn P. van den Heuvel, Robert Boden.

**Software:** Neda Kaboodvand, Behzad Iravani.

**Supervision:** Martijn P. van den Heuvel, Jonas Persson, Robert Boden.

**Validation:** Neda Kaboodvand, Behzad Iravani, Jonas Persson.

**Visualization:** Neda Kaboodvand, Behzad Iravani.

**Writing – original draft:** Neda Kaboodvand.

**Writing – review & editing:** Neda Kaboodvand, Behzad Iravani, Martijn P. van den Heuvel, Jonas Persson, Robert Boden.

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
