## [Decision Letter · Decision Letter 0]

4 Jan 2023

Dear Mr. Iravani,

Thank you very much for submitting your manuscript "Macroscopic resting state model predicts theta burst stimulation response: a randomized trial" for consideration at PLOS Computational Biology.

As with all papers reviewed by the journal, your manuscript was reviewed by members of the editorial board and by several independent reviewers. In light of the reviews (below this email), we would like to invite the resubmission of a significantly-revised version that takes into account the reviewers' comments.

We cannot make any decision about publication until we have seen the revised manuscript and your response to the reviewers' comments. Your revised manuscript is also likely to be sent to reviewers for further evaluation.

Sincerely,

Marcus Kaiser, Ph.D.

Academic Editor

PLOS Computational Biology

Daniele Marinazzo

Section Editor

PLOS Computational Biology

Reviewer's Responses to Questions

**Comments to the Authors:**

Reviewer #1: This is a very interesting and timely study which could be relevant for both basic research and clinical applications. Using whole-brain computational modelling, the authors were identified two depression subtypes exhibited different phenotypic behaviour.

Whereas the study design and rTMS protocol are in line with the standards of depression treatment, the computational modelling seems novel and requires some clarification.

The main concern regarding the results is whether the identified depression subtypes are genuine, and therefore cannot be explained by specific modelling settings.

Technical notes

Page 9, “Filtering was implemented using a discrete cosine transform windowing operation …”. Please specify the window length.

Page 10, Parameter “Cij” (structural connectivity matrix) in equation (1) is not defined/described explicitly. Please add the description.

Page 10, “The variable represents the freely running frequency of region ”. This requires further explanation. How was the running frequency modelled, e.g., frequency range, specific distribution, etc.?

Page 10, “0 denotes the intrinsic frequency at steady state for each region which was experimentally obtained and set to the median frequency of the regional BOLD signal”. Assuming that the power spectrum of a BOLD signal follows a power-law, what does it mean “intrinsic frequency”? To clarify this, could you plot the median frequency (averaged across patients) for each of 68 regions? If possible, could you plot these values on Desikan-Killany brain parcellation?

Page 10, “Plus, an additive Gaussian noise (denoted as ηj, with the standard deviation of β = 0.02) was implemented as Wiener process to simulate random processes in brain”. What is the rationale behind β = 0.02? Have you tested other values?

Page 10, “In order to fit the model to the fMRI data, we need to find an optimal working point, that is a combination of parameters (i.e., A, G, F and M) that maximizes the similarity between the simulated and empirical FC patterns and captures the intrinsic attractor dynamics of the brain”. Have you use any quantitative measure to describe how well the model captures the brain dynamics? Currently, the signals plotted on Fig. 1A do not allow assessing the similarity between simulated and empirical signals. Could you provide an example of simulated and empirical signals from four brain regions (i.e., occipital, parietal, temporal, frontal) of a representative patient, as a supplementary figure?

Page 10, “Accordingly, in a grid-search framework, we searched for parameter A from −0.044 to 0.044 … ”. This could potentially be a strong limitation of the study because the range is arbitrary defined and it seems different from that in Iravani et al., 2021. Could you explain the rationale for this range, and why is it different from your previous study? To make sure that the range of the parameters does not affect the results, could you redo the modelling on a coarser grid: A from −0.2 to 0.2 with a step-size of 0.01, G from 0.005 to 0.05 with a step-size of 0.005, F from 0 to 1 with a step-size of 0.1, and M from 0.1 to 0.5 with a step-size of 0.03.

Page 11, “The correlation values were then converted to z-values …”. Why is it necessary?

Page 11, “The FC maps of half of the patients were randomly selected in each iteration of 1000 Monte Carlo resampling and used to find the optimal parameters for the whole-brain model”. This is an important point. Could you explain why is it important to select half of the patients (not 30% or 70%)? To further validate the modelling technique, could you redo the analysis by selecting (a) 30% and (b) 70% of the patients instead of half?

Page 11, “Applying the probabilistic nonparametric framework enabled us to find the hidden subtypes in the cohort corresponding to different potential local maxima in the parameter distributions”. This requires a detailed description. It remains unclear, what probabilistic nonparametric framework did you use?

Page 11, “Next, based on the similarity between each individual’s empirical and model-derived network measures (i.e., clustering coefficient and nodal strength), we assigned each subject to one of the models (i.e., optimal parameter sets)”. Have you use both measures: clustering coefficient and nodal strength? If so, how did you combine these measures together? It would be interesting to see a table (as supplementary material) where you report these values for each patient (with columns: Patient | Clustering coefficient (empirical, model-derived) | Nodal strength (empirical, model-derived)). It will give more insight regarding the variability of the measures.

Page 11, “Our whole-brain model at the optimal working point …”. Since definition of the optimal point is based on grid-search within a limited range and subsequent correlation between simulated and empirical connectomes, it is difficult to say that this point is truly optimal. It is very possible that there is another working point (associated with different model parameters) that provides slightly lower correlation between the empirical and simulated connectomes, but this point could be equally optimal, if the correlation is not significantly different from the maximum correlation. In your analysis, have you considered other working points with the correlation lower than maximum but above the chance level?

Page 14, “It's worth mentioning that given these two parameter-sets, the model produced regional BOLD-like signals with FC maps resembling the empirical FCs of depression subtypes (rDEP1 = 0.238 and rDEP2 = 0.240 )”. Could you plot these connectomes (i.e., FCs for DEP1 and DEP2) in a supplementary figure? Could you also report the correlation/similarity between these connectomes (i.e., FCs for DEP1 and DEP2)?

Page 16, “Fig. 2”. The main concern here is that there is a large difference between "Sham:DEP2" and "Sham:DEP1, Active:DEP1, Active:DEP2". Could you please plot associated FCs in a supplementary figure? Could you also report the number of subjects in each group (i.e., Sham:DEP2, Sham:DEP1, Active:DEP1, Active:DEP2)?

“Data and Code availability statement”. I would like to ask you to clarify the Data and Code availability statement. It states that “The functional scans, behavioral data and code for analyzing are locally available upon reasonable request”. Could you please provide a reason why the code cannot be made public (upon manuscript acceptance)? Could you also provide information regarding the restrictions associated with distribution of the functional scans and behavioural data?

Thank you!

Reviewer #2: It seems that the results should be reframed according to differential MDD subtype response to placebo/sham rTMS given that this was the group difference driving the interaction between subtype and outcome. This perspective is further supported by the lack of clinical effect (no sig difference for active vs. sham rTMS treatment). The item by item analysis on the symptom scales should be highlighted as exploratory and uncorrected for multiple comparisons which is a major concern for interpreting these findings. The item 8 effects are thus very unlikely to replicate which should be explained clearly in the manuscript. In fact, the result should not be interpreted given the statistical problem and, instead, mentioned only as a preliminary finding that needs to be replicated before it can be confirmed. It should also be mentioned that the subtypes themselves need to be replicated in independent samples to confirm their validity for understanding heterogeneity in clinical response to rTMS at this site.

Minor issues

The Cole et al study did not confirm mechanism (not even pre post fMRI) so the claim that effects happened “via accumulating nonlinear strengthening of synapses” should be removed.

Evidence that the foot MT is better than hand for deep targets should be provided given the claim.

Evidence that this coil reaches thee dmPFC target should be provided.

If the iTBS protocol exactly matches that of the original published (Huang et al.), a reference should be provided. If not, an explanation should be provided.

At 1st mention of the Magventure TMS coil, please provide a model for that coil as they sell many different models.

“To achieve bilateral dmPFC stimulation…we reversed the direction of current” This statement is not supported by the literature. Reversing current does not hop the induced field to the other hemisphere as a rule.

“reaching the predetermined target intensity” Does this mean that the targeted stim level was not reached until quite late in the treatment for some patients? This factor should be accounted for in the analyses since the effective dose would have been different for each patient. Same with the confound of patients who “did not meet these criteria” (presumably never reached the targeted stim level). How often did this happen and what effect did it have on the outcomes?

“identical…protocol…sham” Does this mean that some patients also did not tolerate sham stimulation at the intended target? How did this affect outcomes?

How well did sham work? This is important to determine whether the patients were indeed blind to their condition. This should be quantified in any sham/active comparison study.

For the TMT, what version was used and what are the psychometric properties?

Why rely on structural measures to capture ‘dynamics’ when actual time resolved fMRI data are available?

Why not use individual patient FC to guide targeting since they were collected?

**Have the authors made all data and (if applicable) computational code underlying the findings in their manuscript fully available?**

Reviewer #1: **No: **The authors state that the functional scans, behavioral data and code for analyzing are locally available upon reasonable request.

Reviewer #2: **No: **Their data not shared. They cite/link an online database from other labs only.

PLOS authors have the option to publish the peer review history of their article (what does this mean?). If published, this will include your full peer review and any attached files.

Reviewer #1: No

Reviewer #2: No
---

## [Decision Letter · Decision Letter 1]

16 Feb 2023

Dear Mr. Iravani,

We are pleased to inform you that your manuscript 'Macroscopic resting state model predicts theta burst stimulation response: a randomized trial' has been provisionally accepted for publication in PLOS Computational Biology.

**Please make sure to address the following comment by one of the reviewers when submitting your formatted manuscript: "Most critiques have been addressed. Rather than referring to Boden et al. paper in the critique response, please list the actual details (even if repeated in that manuscript) for the important details missing in the present manuscript."**

Best regards,

Marcus Kaiser, Ph.D.

Academic Editor

PLOS Computational Biology

Daniele Marinazzo

Section Editor

PLOS Computational Biology

Reviewer's Responses to Questions

**Comments to the Authors:**

Reviewer #1: I would like to thank the authors for their response. All my comments have been addressed. Very well done!

Reviewer #2: Most critiques have been addressed. Rather than referring to Boden et al. paper in the critique response, please list the actual details (even if repeated in that manuscript) for the important details missing in the present manuscript.

**Have the authors made all data and (if applicable) computational code underlying the findings in their manuscript fully available?**

Reviewer #1: Yes

Reviewer #2: Yes

PLOS authors have the option to publish the peer review history of their article (what does this mean?). If published, this will include your full peer review and any attached files.

Reviewer #1: No

Reviewer #2: No

---

## [Editor Report · Acceptance letter]

2 Mar 2023

PCOMPBIOL-D-22-01496R1 

Macroscopic resting state model predicts theta burst stimulation response: a randomized trial

Dear Dr Iravani,

I am pleased to inform you that your manuscript has been formally accepted for publication in PLOS Computational Biology. Your manuscript is now with our production department and you will be notified of the publication date in due course.

With kind regards,

Anita Estes
